# Experimental Investigation and Performance Evaluation of Modified Viscoelastic Surfactant (VES) as a New Thickening Fracturing Fluid

**DOI:** 10.3390/polym12071470

**Published:** 2020-06-30

**Authors:** Z. H. Chieng, Mysara Eissa Mohyaldinn, Anas. M. Hassan, Hans Bruining

**Affiliations:** 1Department of Petroleum Engineering, Universiti Teknologi PETRONAS (UTP), Seri Iskandar, Teronoh 32610, Perak, Malaysia; zhinhoung96@gmail.com (Z.H.C.); anas_17005873@utp.edu.my (A.M.H.); 2Civil Engineering and Geosciences, Delft University of Technology (TU-Delft), Stevinweg 1, 2628 CE Delft, The Netherlands; a.m.hassn@tudelft.nl

**Keywords:** sand suspension, temperature resistance, viscoelastic surfactant (VES), gel-breaking, core-damage

## Abstract

In hydraulic fracturing, fracturing fluids are used to create fractures in a hydrocarbon reservoir throughout transported proppant into the fractures. The application of many fields proves that conventional fracturing fluid has the disadvantages of residue(s), which causes serious clogging of the reservoir’s formations and, thus, leads to reduce the permeability in these hydrocarbon reservoirs. The development of clean (and cost-effective) fracturing fluid is a main driver of the hydraulic fracturing process. Presently, viscoelastic surfactant (VES)-fluid is one of the most widely used fracturing fluids in the hydraulic fracturing development of unconventional reservoirs, due to its non-residue(s) characteristics. However, conventional single-chain VES-fluid has a low temperature and shear resistance. In this study, two modified VES-fluid are developed as new thickening fracturing fluids, which consist of more single-chain coupled by hydrotropes (i.e., ionic organic salts) through non-covalent interaction. This new development is achieved by the formulation of mixing long chain cationic surfactant cetyltrimethylammonium bromide (CTAB) with organic acids, which are citric acid (CA) and maleic acid (MA) at a molar ratio of (3:1) and (2:1), respectively. As an innovative approach CTAB and CA are combined to obtain a solution (i.e., CTAB-based VES-fluid) with optimal properties for fracturing and this behaviour of the CTAB-based VES-fluid is experimentally corroborated. A rheometer was used to evaluate the visco-elasticity and shear rate & temperature resistance, while sand-carrying suspension capability was investigated by measuring the settling velocity of the transported proppant in the fluid. Moreover, the gel breaking capability was investigated by determining the viscosity of broken VES-fluid after mixing with ethanol, and the degree of core damage (i.e., permeability performance) caused by VES-fluid was evaluated while using core-flooding test. The experimental results show that, at pH-value (6.17), 30 (mM) VES-fluid (i.e., CTAB-CA) possesses the highest visco-elasticity as the apparent viscosity at zero shear-rate reached nearly to 10${}^{6}$ (mPa·s). Moreover, the apparent viscosity of the 30 (mM) CTAB-CA VES-fluid remains 60 (mPa·s) at (90 ${}^{\circ }$C) and 170 (s${}^{-1}$) after shearing for 2-h, indicating that CTAB-CA fluid has excellent temperature and shear resistance. Furthermore, excellent sand suspension and gel breaking ability of 30 (mM) CTAB-CA VES-fluid at 90 (${}^{\circ }$C) was shown; as the sand suspension velocity is 1.67 (mm/s) and complete gel breaking was achieved within 2 h after mixing with the ethanol at the ratio of 10:1. The core flooding experiments indicate that the core damage rate caused by the CTAB-CA VES-fluid is (7.99%), which indicate that it does not cause much damage. Based on the experimental results, it is expected that CTAB-CA VES-fluid under high-temperature will make the proposed new VES-fluid an attractive thickening fracturing fluid.

## 1. Introduction

Hydraulic fracturing is a technique that is applied for more than 40 years in the petroleum industry [1,2] and, nowadays, plays an important role in hydrocarbon recovery in unconventional reservoirs [3,4]. Thanks to the recent development of unconventional hydrocarbon recovery methods, hydraulic fracturing can be used to improve the productivity of the hydrocarbon reservoirs [5,6] in various field-applications, including shale gas [7,8]. Hydraulic fracturing involves the injection of a high pressure fracturing fluid into a reservoir formation [9,10] with the purpose to create a fracture, and carrying a transported proppant into the fracture [11,12] and, thus, with this way one can maintain a high formation conductivity. [13,14]. At an early stage of hydraulic fracturing, polymer fluids such as guar gum [15,16] was used mostly as a thickening agent for the fracturing fluid [17,18]. Laboratory experiments have shown that traditional polymer-based fluids generate residue(s), which can damage the formation and reduce the pore conductivity. As reported by Thomas et al. [19,20], only (30–45%) of the injected guar-based polymer fluids could return from the well during the flow-back period as the unbroken residues from the polymer-based fracturing fluid can plug the flow path [21,22,23]. Due to the poor sand suspension ability, the proppant will settle at the bottom of the polymer fluids before reaching the fracture tip. The high viscosity nature exhibited by polymer fluids could extend the fracture in height instead of in length [24,25,26]. Therefore, the concept of VES-fluid as thickening agent for the fracturing fluid has been introduced by Schlumberger in 1997 [27,28,29]. The VES-fluid had attracted attention due to the viscoelastic behaviour and the nature of the non-residue(s) generated after gel breaking [29,30,31]. The sand suspension ability of Visco-elastic Surfactant (VES) fluid depends on the elasticity instead of the viscosity [27]. This viscoelastic behaviour is attributed by the formation of wormlike micelles (WLMs) [28]. In addition, smart wormlike micelles (WLMs) or smart self-organized structures, can be used in a myriad of applications within the oil and gas industry, including hydraulic fracturing, emulsions, polymer, surfactant, and foam flooding [15,32]. Nowadays, investigation is carried out to develop WLMs that respond to external stimuli for the ease to control visco-elasticity [33,34]. The VES fluids are known as the clean fracturing fluids as the fluid does not generate any residue after gel breaking, which can reduce the formation damage [20,35]. VES-fluid has observable benefits, such as easy preparation, good sand suspension ability, extremely low damage to reservoir formation, as compared to injected polymer-based fluids. Despite all the advantages of conventional VES fluids, investigation shows that conventional VES fluids exhibit low thermal and shear resistance in deeper oil and gas reservoirs [36,37,38]. The deficiency is due to the single-chain surfactant acquired by the conventional VES fluid. Apart from that, the conventional single-chain VES fluids shown in Figure 1 have a high critical micelle concentration (CMC), at which high concentrations of surfactants are required to form a high viscoelastic fracturing fluid and, thus, this would increase the cost [39,40,41].

In this work, two modified VES-fluid are developed as new thickening fracturing fluids, which consist of more single-chain coupled by hydrotropes (i.e., ionic organic salts) through non-covalent interaction. The VES-fluid composed of two single-chain (CTAB) surfactants connected by a spacer group is known as Gemini-like VES-fluid (see Figure 2, while the trimer VES-fluid consists of three single-chain (CTAB) surfactants that are connected by spacer groups. The spacer group can restrict the electrostatic repulsion between (i.e., amphoteric, which consists of hydrophilic and hydrophobic) groups in order to increase the density of the surfactant (head) groups [42,43,44]. A series of experimental performance tests were carried out in order to evaluate the new modified CTAB-based VES-fluids. The performance evaluation tests are include (1) the rheological behaviour, (2) temperature and shear resistance, (3) sand suspension ability and gel breaking capability, and (4) core flooding tests. Finally, comparison was made between the performance of a conventional VES-fluid and the two (new) modified VES-fluids as potential thickening fracturing fluids. The main objective of this contribution is to carry out the experimental investigation and evaluate the performance of CTAB-based VES-fluid as a potential new thickening fracturing fluid.

## 2. Materials and Methods

### 2.1. Materials

The Cationic Surfactant, i.e., Cetyltrimethylammonium bromide (CTAB), citric acid (CA), and maleic acid (MA) (see Figure 3 and Figure 4) were obtained from Avantis Laboratory Supply (Ipoh-Perak, Malaysia). Hydrochloric acid and sodium hydroxide were purchased from Benue Sains Sdn Bhd (BSSB) Co., Ltd. (Kuala-Lumpur, Malaysia). Ethanol and proppant were obtained from the laboratory of the Universiti Teknologi Petronas (UTP) in Malaysia. In all experiments, the water was triply distilled using the quartz-water purification method. Moreover, all reagents are used without further treatment.

### 2.2. Methods

Surfactants are among the most versatile amphoteric organic substances of the chemical industry, appearing in such diverse products in the oil and gas applications [21,22,23].The cationic-surfactant self-assembly and pH-responsive (pH-dependent) CTAB in aqueous solutions is an obliging phenomenon and entropy driven process primarily governed by hydrophobic interaction [45,46]. Moreover, the addition of the surfactant CTAB to VES-fluid solution results in increased viscosity at increase (low to moderate) surfactant concentration, where the surfactant binds the hydrophobic groups together [46]. The shape and size of ionic-surfactant micelles depend on polar head group, hydrocarbon tail, and counter ion in the surfactant based formulation [47,48]. Micellar transition in an aqueous solution is a technologically and scientifically significant area. Spherical micelles form at concentration, above the critical micelle concentration (CMC), and they may grow to rod-like or worm-like structures (WLS). The spherical, ellipsoidal, rod, or worm-like structures may also depend on the presence of additives [45]. CTAB forms spherical micelles having a diameter (2 to 3) (nm) [49,50], CMC equal to 1 (mM) [45], and a krafft point around 25 [51] in water. These micelles undergo sphere-to-rod transition above 250 (mM) [52,53]. In this work, the micellar transition in CTAB solutions (i.e., CTAB-AC and CTAB-MA based VES-fluid) has been studied by viscosity (i.e., viscoelastic behaviour) [54,55].

#### 2.2.1. Samples Preparation

The modified VES fluids were prepared by mixing the cationic surfactant CTAB with organic acids, citric acid or CA (i.e., weak acid that is formed in the tricarboxylic acid cycle (C${}_{6}$H${}_{8}$O${}_{7}$)) and maleic acid or MA (i.e., organic compound that is a dicarboxylic acid (CHCO${}_{2}$H)) at the molar-ratio of 3:1 and 2:1, respectively. A formula was used to calculate the amount of chemical (gram) and the weighing process was carried out by using an electronic balance. Mechanical or magnetic stirring is used to mix the measured amount of chemical and dissolve it completely in the solution. Apart from that, the solution is required to be sealed with transparent plastic to prevent the evaporation process [56,57]. The pH-value of the VES-fluid was attuned to the desired pH-value by adding NaOH or HCl to the solution and then measured using a pH-meter or litmus paper. More acidic conditions were made by adding HCl (aqueous), and the alkaline range of pH-values were achieved by adding of (NaOH). The prepared samples were left at ambient temperature (25 ${}^{\circ }$C) for one day. For convenience purposes, throughout this paper, the desired concentration of CTAB mixed with citric acid (CA) and maleic acid (MA) will be referred to CTAB-MA and CTAB-CA, respectively. Additionally, for simplicity, VES-fluid produced by 30 (mM) CTAB and 15 (mM) CA will be denoted as the 30 (mM) CTAB-CA VES-fluid.

#### 2.2.2. Performance Evaluation Experiments

#### 2.2.3. The Dynamic and Steady Rheological Tests

Because smart wormlike micelles (WLMs) formed by the VES-fluids are affected by the pH values, 30 (mM) VES fluids with a pH ranging from 2 to 9 (2 < pH < 9) were prepared. The steady and dynamic rheological behaviour of different VES-fluids as function of pH were tested using the Physica (MCR-301) rheometer at (25 ${}^{\circ }$C). By analyzing the relationship between the shear-rate and viscosity for the VES-fluids at different pH-values, the optimum pH-range for the highest viscoelasticity value(s) was determined.

#### 2.2.4. Temperature and Shear Resistance Test

Using the rheometer (Physica MCR-301), different types of VES-fluids were poured into the rotor cup and heated slowly starting from (30 ${}^{\circ }$C) with a heating rate of 3 (${}^{\circ }$C/min.), the rotator spin-speed was fixed at shear-rate of 170 (s${}^{-1}$). After reaching a temperature of (90 ${}^{\circ }$), the temperature and shear rate were maintained for two h. Finally, the apparent-viscosity of the tested fracturing-fluid at each temperature was recorded. By using the rheometer again, the VES-fluids were tested for their ability to maintain the apparent viscosity after undergoing an alternate shear process. The rotator spins at the alternate shear rate of 50 (s${}^{-1}$), 100 (s${}^{-1}$), and 150 (s${}^{-1}$) at a constant temperature of (25 ${}^{\circ }$C).

#### 2.2.5. Sand Suspension Ability Test

Sand proppant was put into VES-fluid (i.e., CTAB-CA and CTAB-MA) slowly and the time (seconds) for the sand proppant to suspend until reaching the target point was recorded. Figure 5 is used to calculate the sand suspension velocity (mm/s). The sand proppant used is standardized, which could pass through the 40-mesh screen. The experiment was repeated using the two types of Visco-elastic Surfactant VES-fluids (CTAB-CA and CTAB-MA) (see Figure 6. Thermostat water bath was used to heat the VES-fluids to higher temperature to simulate the reservoir condition (i.e., 90 ${}^{\circ }$C).

#### 2.2.6. Gel Breaking Capability Test

Ethanol with concentrations of (5%, 10%, and 15%) was added separately to the VES-fluids (i.e., CTAB, CTAB-CA, and CTAB-MA) under specified temperatures (i.e., 30 ${}^{\circ }$C and 90 ${}^{\circ }$C) and the solution was stirred. The ethanol added is used to decrease the viscosity of the VES fluid, so that it does not cause any damage to the formation after treatment. The upper part of the mixture was tested by using a viscometer within a certain time range until the mixture was broken. The VES-fluid is considered to be broken when the viscosity is lower than 5 (mPa·s). The time that was taken for the mixture to broke was recorded. The surface tension and the interfacial tension of the mixture were tested using a tensiometer. The experiment was repeated using different types of VES-fluids (i.e., CTAB, CTAB-CA, and CTAB-MA).

#### 2.2.7. Core damage Evaluation Test

Before measuring the length, diameter, and the weight of Indiana Limestone core sample, the core sample was cleaned by using the Dean Stark Machine, and then dried in the oven under (90 ${}^{\circ }$C) for half a day. The porosity and permeability of dry core sample were then measured using the gas permeameter. Next, the core sample was saturated with brine for 12-h using a desiccator. After saturation with brine, the core flooding test was performed at (10 ${}^{\circ }$C) and at (70 ${}^{\circ }$C). The core sample was flooded first with brine (until the pressure difference is stable) for the purpose of calculating the core sample permeability after brine injection. By using the Darcy formulation, the permeability can be calculated from the different pressure points. 30 (mM) CTAB-CA VES-fluid was then injected (by the high-pressure pump) into the core until pressure-drop was observed showing the fluid breakthrough. The core sample was flushed with brine-solution to remove the stimulation fluid, and it was cleaned, dried, and measured (e.g., porosity and permeability measurements).

## 3. Results and Discussion

### 3.1. Phase Behavior

The pH directly affects the electrostatic interactions forces (i.e., double layer repulsion and van der Waals attraction) and hydrogen bonds between the cationic surfactant and organic acid by changing their molecular state distributions [58,59]. Therefore, it is crucial to investigate the effect of pH on the phase behaviour of 30 (mM) CTAB-CA and 30 (mM) CTAB-MA fluids by observing their macroscopic appearance changes [60,61,62]. Macroscopic appearance photos of each CTAB-CA VES-fluid and CTAB-MA VES-fluid (at various pH-values and at 25 ${}^{\circ }$C) are shown in Figure 7 and Figure 8, respectively. Figure 7 shows that CTAB-CA VES-fluid changes from transparent to cloudy solution at (pH = 7.00), while Figure 8 shows that CTAB-CA VES-fluid changes from transparent to cloudy solution at (pH = 7.15). We can conclude that both of the solutions appear transparent at (pH < 7), while it started turning cloudy at (pH > 7).

### 3.2. Rheological Properties

#### 3.2.1. The Effect of pH on Dynamic and Steady Rheological Tests

Figure 9 shows the dynamic rheological curves for CTAB-CA (i.e., 30 mM CTAB + 15 mM CA) VES-fluid at different pH-values. In Figure 9 storage modulus (G’), and the loss modulus (G”) are varying as a function of oscillatory shear-frequency for different pH-values. The higher the pH-value, the more the gap between (G’) and (G”) and, hence, the more time is needed from the fluid to lose its viscoelasticity and turn into viscous fluid. On the other hand, the gap between the storage modulus (G’) and loss modulus (G”) is getting narrower with the increase in shear-frequency sweep. From the degree of variation at the considered pH-values (3 < pH < 7), it is expected that the CTAB-CA VES-fluid may turn from viscoelastic behaviour to viscous behaviour at shear frequency higher than 100 (rad/s). The fastest of this change of behaviour (i.e., turning from viscoelastic to viscous) will be at pH (3.06), as the gap is getting narrower the most. If we can assume that the curves of storage modulus (G’) and loss modulus (G”) are extrapolatable, then we expect that the two curves of (G’) and (G”) for pH-value (3.06) will intersect at shear-frequency less than 100 (rad/s). The other observation that can be pointed out from the Figure 9 is that the effect of pH can induce different micelles structures and, thus, adjust the viscoelasticity properties of CTAB-CA VES-fluid between liquid like and solid-like and [63]. This classification can be recognized from the dependency of the storage modulus (G’) and loss modulus (G”) on the shear frequency. If the storage modulus (G’) and loss modulus (G”) are independent of the shear-frequency, then the VES-fluid is considered solid-like, whereas if (G’) and (G”) are dependent of the shear-frequency, then the VES-fluid is considered liquid-like. Furthermore, at pH-value of (3.03), the storage modulus (G’) and loss modulus(G”) increase slightly with the increase of the shear frequency. Consequently, the VES-fluid shows a liquid-like behaviour. For pH value of 4.04 the storage modulus exhibits a shear-frequency dependence behaviour at low shear-frequency (up to 3 rad/s), and then it becomes independent of the shear-frequency. At pH (6.17) the VES-fluid behaves as solid-like (i.e., well structured), which indicates that the VES-fluid is stable. In summary, the CTAB-CA VES-fluid shows a viscoelastic property at all pH-values in acidic range (pH < 7). The change of pH can induce different micelles structures and, thus, modify viscoelasticity properties of CTAB-CA VES-fluid between liquid like (i.e., pH = 3.03) and solid-like (i.e., pH = 6.17).

In the same way as in Figure 9 and Figure 10 shows the dynamic rheological curves for CTAB-MA (i.e., 30 mM CTAB + 15 mM MA) VES-fluid at different pH-values. In Figure 10, storage modulus (G’), and the loss modulus (G”) are changing as a function of oscillatory shear-frequency for various pH-values (4 < pH < 7) at 25 ${}^{\circ }$C. In general, the dynamic-rheological behaviour of the CTAB-MA VES-fluid shows both viscoelastic (i.e., G’ > G”) and viscous (i.e., G’ < G”) behaviour at different shear-frequency. The critical shear-frequency value above which the VES-fluid exhibits viscoelastic behaviour is pH-dependent i.e., the higher the pH value, the lower is the critical shear-frequency value. For the pH-values of (6.17), the CTAB-MA VES-fluid turns from a viscous to viscoelastic behaviour at shear-frequency of 0.1 (rad/s). Whereas, for pH-value of (4.02), the VES-fluid shows a viscoelastic behaviour that is above critical shear-frequency value of 1 (rad/s). The relaxation time of 30 mM CTAB-MA VES-fluid increases as the pH-values of the solution increased from (4.02) to (6.17). Moreover, looking at the dependency of storage modulus on the shear-frequency, it can be observed that the CTAB-MA VES-fluid exhibits liquid-like behaviour at all pH-values in the acidic range (4 < pH < 7) at (25 ${}^{\circ }$C).

Figure 11 shows the steady rheological curve for CTAB-CA (i.e., 30 mM CTAB + 15 mM CA) VES-fluid at different pH-values. In Figure 11, the apparent viscosity is varying as a function of oscillatory shear-frequency for different pH-values (3 < pH < 10) at 25 ${}^{\circ }$C. At the acidic range of pH (3 < pH < 7), we observe that CTAB-CA VES-fluid exhibits non-Newtonian shear-dependent (shear-thinning) behaviour, which is indicated by the reduction of the viscosity while the shear rate increases. On the other hand, at the alkaline range of pH (i.e., pH = 8.05 and pH = 9.15), the CTAB-CA VES-fluid shows Newtonian fluid behaviour, where its viscosity becomes independent of the shear rate. In addition to the effect of shear rate, the viscosity is also affected by the pH. At acidic range (3 < pH < 7), the viscosity of the CTAB-CA VES-fluid increases as the pH-value increases. Whereas, in the alkaline range (i.e., pH = 8.05, and pH = 9.15), the viscosity of the CTAB-CA VES-fluid decreases, while the pH-value increases. Note that the viscosity of CTAB-CA VES-fluid reaches its highest value (106 mPa·s) at the pH value of (6.17).

In the same way, Figure 12 shows the steady rheological curve for CTAB-MA (i.e., 30 mM CTAB + 15 mM) VES-fluid at different pH-values. In Figure 12, the apparent viscosity is changing as a function of oscillatory shear-frequency for different pH-values (2 < pH < 8) at 25C. At the acidic range of pH (2 < pH < 7), we see that CTAB-MA VES-fluid shows non-Newtonian shear-dependent (shear-thinning) behaviour, which is indicated by the reduction of the viscosity, while the shear rate increases. This shear dependency behaviour is more clear above the critical shear rate value, where the apparent viscosity starts to decrease. However, at the alkaline range of pH (i.e., pH = 7.33, and pH = 7.99), the CTAB-MA VES-fluid shows Newtonian fluid behaviour, where its viscosity becomes independent of shear rate. In addition to the effect of shear rate, the viscosity is also affected by the pH. At acidic range (i.e., 2 < pH < 7), the viscosity increases as the pH increases. At alkaline range (i.e., pH = 7.33 and pH = 7.99), the viscosity of the CTAB-MA decreases while the pH increases. The viscosity of CTAB-MA VES fluid reaches its highest value (104 mPa·s) at the pH-value of (7.00).

Figure 13 shows the apparent viscosity for the three types of CTAB based (CTAB, CTAB-MA, and CTAB-CA) VES-fluid at different pH-values and at constant low shear rate. Figure 13 indicates that, among the three CTAB-based VES-fluids, CTAB-CA VES-fluid has the best performance due to its highest viscosity, which is an important property that needs to be obtained during hydraulic fracturing process. Furthermore, the graph also indicates that the optimum pH-value(s) is in the range between (6 and 7), which need to be maintained for a proper fracturing fluid performance. Note that, during hydraulic fracturing, the fluid rheology also affects the pumpability of the VES-fluid as higher viscosity leads to higher pressure losses. Therefore, the optimum selected viscosity value should be decided based on the availability of the pumping facilities. In addition, it should be considered that during injection, the viscosity will decline substantially with a rate that is proportional to the degree of shear rate, as indicated in the viscosity versus shear rate results (i.e., see Figure 14).

#### 3.2.2. Combining the Effect of pH on Rheological Tests Results with Phase-Behaviour Study

The effect of pH on the apparent viscosity for the three types of CTAB based (CTAB, CTAB-MA, and CTAB-CA) VES-fluids (i.e., at different pH-values, at constant low shear rate, and at 25 ${}^{\circ }$C) was studied and the results are shown in Figure 13. At an acidic pH range, the apparent viscosity of the three VES-fluids increases slowly with the pH until the pH-value reaches less than 7 (i.e., around pH = 6.17). With a further increase of pH after the pick viscosity (pH > 7), the phase separation gradually occurred in both CTAB-CA and CTAB-MA VES-fluid (i.e., solutions become non-transparent), which results in a significant drop of viscosity (see Figure 13 and Figure 14). The phase separation behaviour for CTAB-CA and CTAB-MA VES-fluid can be observed in Figure 7 and Figure 8, respectively. Moreover, combining the effect of pH on rheological (i.e., dynamic and Steady) (see Figure 9, Figure 10, Figure 11 and Figure 12) and apparent viscosity (see Figure 13 and Figure 14) tests results with the phase behaviour results of CTAB-CA and CTAB-MA based VES-fluid (i.e., see Figure 7 and Figure 8), we can assert that it is recommended to avoid creating alkaline environment (pH > 7) during the hydraulic fracturing operation. This is not only to avoid low viscosity, but also to prevent the near-wellbore-region formation from being damaged due to (VES-fluid) precipitation.

In addition, the results apparent-viscosity pH dependency of VES-fluids were combined with the results that were reported by Wang and co-workers, Zhang et al., and Wanli kang et al. [21,43,63]. Figure 15 indicates that the variation of viscosity with pH of our VES-fluids i.e., CTAB, CTAB-CA, and CTAB-MA are following similar trends of (UC22AMPM/PPA) and (CTAB/PPA) reported by Wang et al. [43,63]. For all of the five VES-fluids, viscosity increases with pH increases up to a maximum point and then starts to decrease with pH increase. The critical pH-value above which the viscosity starts to decrease falls within the pH-range of (6-7) except for CTAB/PPA, where the critical point falls in high acidic range of pH (2< pH <3). Additionally, (CTAB/PPA) shows pH irresponsive behaviour in the alkaline range of pH. It can be observed that both of our CTAB-based VES-fluids (i.e., CTAB-MA and CTAB-CA) perform better than (CTAB/PPA). In addition, our (CTAB-CA) shows almost similar performance as that of (UC22AMPM). As pointed out by Wanli Kang et al. [63], (UC22AMPM) processes a higher viscosity than (CTAB) because of the difference in the hydrophobic chain length, which affects the packing parameter of surfactant molecules and, hence, the length of wormlike micelles. Based on that, we can explain the micro structure changes that made (CTAB-CA) to possess higher viscosity and more viscoelasticity tendency as compared to that of (CTAB-MA). We believe that CTAB-CA has got longer hydrophobic chains than those of (CTAB-MA), which enhanced the packing parameter of surfactant molecules and (WLM).

### 3.3. Temperature and Shear Resistance

The thermal stability tests were carried out at (90 ${}^{\circ }$C) and at the constant shear rate of 170 for 120 min. to investigate the temperature resistance and shear stability of the modified 30 (mM) CTAB-CA and CTAB-MA as well as the conventional 30 (mM) CTAB-based VES-fluids. The increasing of temperature leads to make the inter-molecular force weak and it breaks the 3D network structure of WLMs [64]. Figure 16 shows that the apparent viscosity of the three types of CTAB based (CTAB, CTAB-MA, and CTAB-CA) VES-fluids decrease rapidly with increasing the shear-time duration. This indicates that all three types of VES-fluids exhibit psedoelastic (viscoelastic) shear-dependent fluids but with different degrees of shear dependency. Additionally, we observe that, the shear-dependency behaviour continues with increasing the shear-time duration until reaching a critical shear time, after which the three CTAB-based VES-fluids exhibit shear-independent fluid behaviour, while the viscosity remains constant. Moreover, the apparent viscosities declining-rate of the 30 (mM) CTAB VES-fluid is the highest, which was followed by the 30 (mM) CTAB-MA VES-fluid and 30 (mM) CTAB-CA VES-fluid. After 120 min. shear-time and at (90 ${}^{\circ }$C), the final apparent viscosity of the 30 mM CTAB, 30 (mM) CTAB-MA, and 30 (mM) CTAB-CA VES-fluids are maintained around 35 (mPa·s), 40 (mPa·s), and 50 mPa·s, respectively. The apparent viscosity of 30 (mM) CTAB-CA fluid reaches the highest value, which could be concluded to acquire the best temperature and shear resistance performances. According to the operating standard, all the fluids with the apparent viscosity higher that 25 (mPa·s) at high temperature satisfies the requirement as a thickening fracturing fluid [64]. The other significant point to be highlighted in Figure 16 is the non-uniform variation of viscosity versus shear-time during the heat stability test of the 30 (mM) CTAB-MA VES-fluid. At the beginning, the viscosity of the 30 (mM) CTAB-MA VES-fluid declines with the shear-time from 60 (mPas) to 56 (mPa·s) after 7 min. of shear-time, and then viscosity increases sharply to 75 (mPa·s) at a shearing time of 12.5 min.; and, the end it follows the same (liner) trends that followed by 30 (mM) CTAB and 30 (mM) CTAB-CA VES-fluids. Note that the same behaviour has been reported by J. Zhao and coworkers for the (5 wt %) VES-T and 1.4 (wt %) KCl fracturing fluid [23].

### 3.4. Sand Suspension Ability

A fracturing fluid with good proppant suspension properties has the ability to carry proppant into the cracks equally with a higher sand ratio, and it can also carry a larger size proppant. The proppant would suspend quickly in the fracturing fluid if the sand suspension ability is weak. Because the dynamic sand suspension test is quite complicated [65,66], the static testing method was applied in this study (see Figure 5). The tests were carried out with different VES-fluids under (30 ${}^{\circ }$C) and (90 ${}^{\circ }$C). The results show that, under (30 ${}^{\circ }$C), CTAB-CA, CTAB-MA, and CTAB VES fluids showed a stable proppant suspension property over the time range considered, as depicted in Table 1. The proppant does not suspend and remains above the VES-fluids even after a few days due to the high viscosity of VES-fluids at normal temperature (i.e., 30 ${}^{\circ }$C). The sedimentation velocities of sand proppant for 30 (mM) CTAB-CA, CTAB-MA, and CTAB under (90 ${}^{\circ }$C) are 4.37 (mm/s), 2.72 (mm/s), and 1.67 mm/s, respectively. The highest apparent viscosity of CTAB-CA fluid allows the longest sand suspension time and the slowest average suspension velocity. From the results of the sand suspension ability test, we can see that the average velocity of the sand increases as the temperature increases, due to the lower viscosity of the fluid at higher temperature. In addition, the sand suspension ability of the fracturing fluids becomes weaker as the time goes on due to the presence of a breaker additive. According to the operation standard, the acceptable sand suspension velocity for the VES fracturing fluid is in the range between 0.5 (mm/s) and 5.0 (mm/s), indicating that all of the VES-fluids can achieve the criteria as a thickening fracturing fluid to suspend the proppant.

### 3.5. Gel Breaking Capability

After treatment, the VES fracturing fluid must break immediately and completely to prevent any formation damage. According to the standard operation for the VES fracturing fluid, the viscosity of the broken fluid is below 5 mPa·s [67]. For the purpose of breaking the gel, the VES-fluid must be in contact with either brine solutions or hydrocarbon [68,69]. In this case, ethanol was added to the VES fluids with different ratio’s and the changes in the viscosity of mixture were investigated after a certain time range. The experiments were done under (30 ${}^{\circ }$C) and (90 ${}^{\circ }$C), respectively.

30 mM CTAB-CA, CTAB-MA, CTAB VES-fluids will break faster when the ethanol to fluid ratio is higher, which causes the lower viscosity after breaking. For the ethanol–VES fluid ratios of (1:20) and (1:10), the gel-breaking did not occur under (30 ${}^{\circ }$C) and (80 ${}^{\circ }$C), even after 24 h. However, for the ethanol–VES fluid ratio of (15:100), all 30 (mM) VES-fluids are completely broken at (30 ${}^{\circ }$C) and at (80 ${}^{\circ }$C); and the viscosities of both broken solutions are less than 5 (mPa·s) at duration of 120 min. As shown in Table 2, the breaking time for 30 (mM) CTAB-CA VES-fluid is the longest at (30 ${}^{\circ }$C) and at (80 ${}^{\circ }$C), followed by 30 (mM) CTAB-MA and 30 (mM) CTAB. This is due to the highest viscosity of CTAB-CA fluid, thus a longer time to break. It can be observed that the higher the temperature, the quicker the solution breaks. This is due to the low visco-elasticity of the fracturing fluids under high temperature. Moreover, the surface tension and the interfacial tension of all broken VES-fluids (i.e., see Table 3 are achieving the operating standard of the fracturing fluid at which they are lower than that of the broken conventional fracturing fluids. In summary, all of the VES fluids acquire good gel breaking capability.

### 3.6. Core Flooding Evaluation Test

CTAB-CA VES fluid was chosen to test its core damage properties. The porosity and permeability of the core sample before and after core-flood test were measured by using a gas permeameter. Table 4 shows the core properties pf the VES-fluid before and after the core-flood test. The core damage rate (Cd) of the CTAB-CA gel-breaking fluid is shown in Figure 17. The initial permeability ($K_{0}$) of the dry core sample, before the core-flood test, is 189.93 mD, as shown in Table 4. The injected pressure has increased after injecting the CTAB-CA fluid and the permeability ($K_{1}$) is dropped to 63.31 (mD). The brine was then injected to clean up the VES-fluid inside the core. The final permeability ($K_{2}$) of the core-sample (after drying) is 174.74 (mD). The final permeability of the core is slightly reduced when compared to the initial permeability due to a small amount of core damage. The core damage rate of the CTAB-CA VES fluid is (7.99%) [36], which is lower than the standard core damage rate of 20%. In summary, based on the core-flood test, CTAB-CA VES-fluid does not cause much damage to the formation.

## 4. Conclusions

To improve the performance of a conventional single-chain VES fluid as thickening agent for fracturing fluid, modified VES fluids were introduced. In this work, two types of modified VES fluids were developed as new thickening fracturing fluids through the formulation of mixing long chain cationic surfactant cetyltrimethylammonium bromide (CTAB) with organic acids, which are citric acid (CA) and maleic acid (MA) at a molar ratios of (3:1) and (2:1), respectively. As an innovative approach, CTAB and CA are combined to obtain a solution (i.e., CTAB-based VES-fluid) with optimal properties for fracturing and this behaviour of the CTAB-based VES-fluid is experimentally corroborated. From the The experimental results, we can summarize and conclude the following:The dynamic and steady state rheological investigation of CTAB-CA and CTAB-MA VES-fluids showed that the synthesized VES fluids mostly exhibit shear-dependant viscoelastic non-Newtonian behaviour.In general, both CTAB-CA and CTAB-MA VES-fluids are found to be pH responsive at both acidic and alkaline conditions.
The effect of the pH shows higher viscosity and more structured solid-like behaviour at acidic range of pH (i.e., pH < 7).Whereas, at an alkaline range of pH (i.e., pH > 7), the effect of the pH shows lower viscosity and more liquid-like behaviour is observed.CTAB-CA VES-fluid shows more viscoelasticity and solid-like nature compared to CTAB-MA.
At 25 The viscosity of CTAB-CA can reach up to 106 mPa.s at pH-value of 6.17, whereas the maximum viscosity of CTAB-MA is 10${}^{4}$ mPa.s.CTAB-CA VES-fluid has the best temperature and shear resistance, its apparent viscosity remains at 65 mPa.s; after continuous shearing for 2 h at 90 ${}^{\circ }$C and shear rate of 170 (s${}^{-1}$).CTAB-CA VES-fluid exhibits excellent sand suspension and gel breaking ability; At 90 oc, the sand suspension velocity of CTAB-CA was found to be 1.67 mm/s and complete gel breaking was achieved within 2 h after mixing with the ethanol at the ratio of 10:1.The core flooding tests show that, after injecting CTAB-CA fluid into the core sample, the core damage rate is 7.99%, indicating that it doesn’t cause much damage.Based on the performance evaluation results, it is expected that CTAB-CA VES-fluid under high-temperature will make the proposed new VES-fluid an attractive thickening fracturing fluid.

## Figures and Tables

**Figure 1 polymers-12-01470-f001:**
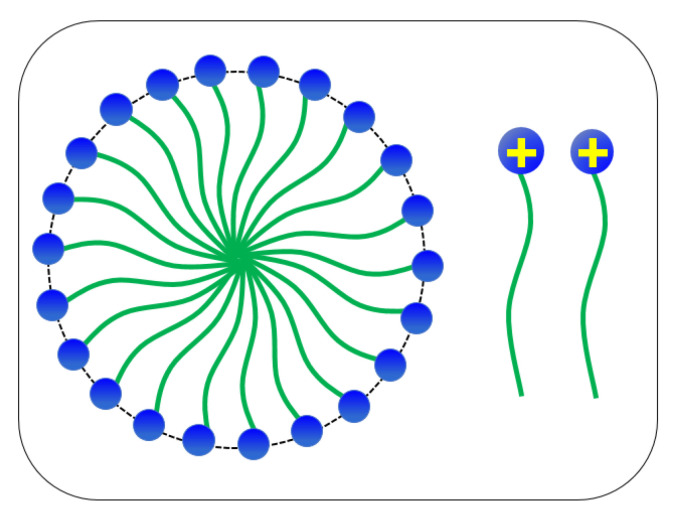
Conventional single-chain VES.

**Figure 2 polymers-12-01470-f002:**
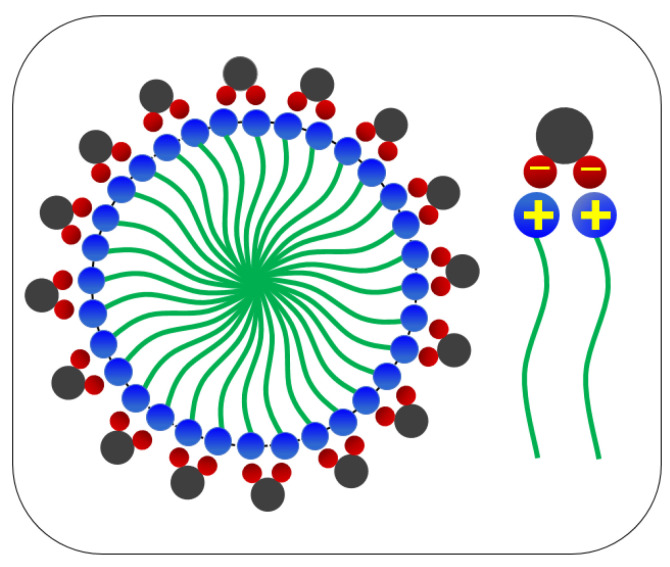
Modified Gemini-like Visco-Elastic Surfactant (VES).

**Figure 3 polymers-12-01470-f003:**
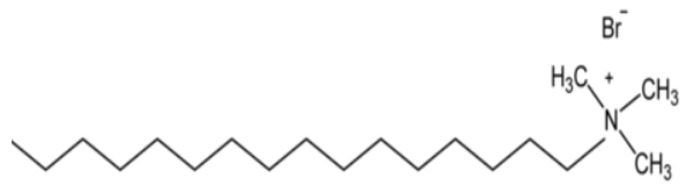
Chemical structure of CTAB.

**Figure 4 polymers-12-01470-f004:**
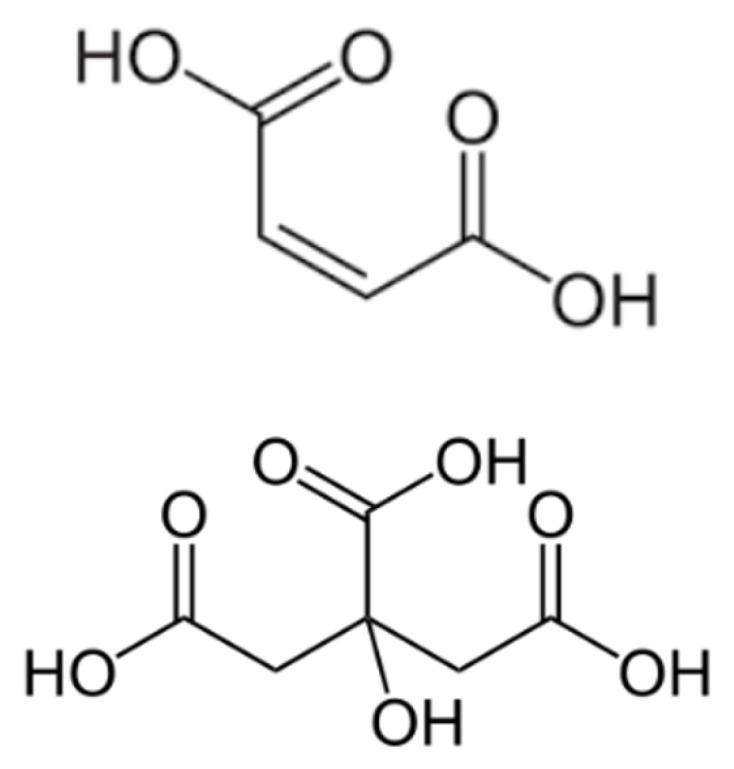
Chemical structure of Citric-acid, and Maletic-acid.

**Figure 5 polymers-12-01470-f005:**
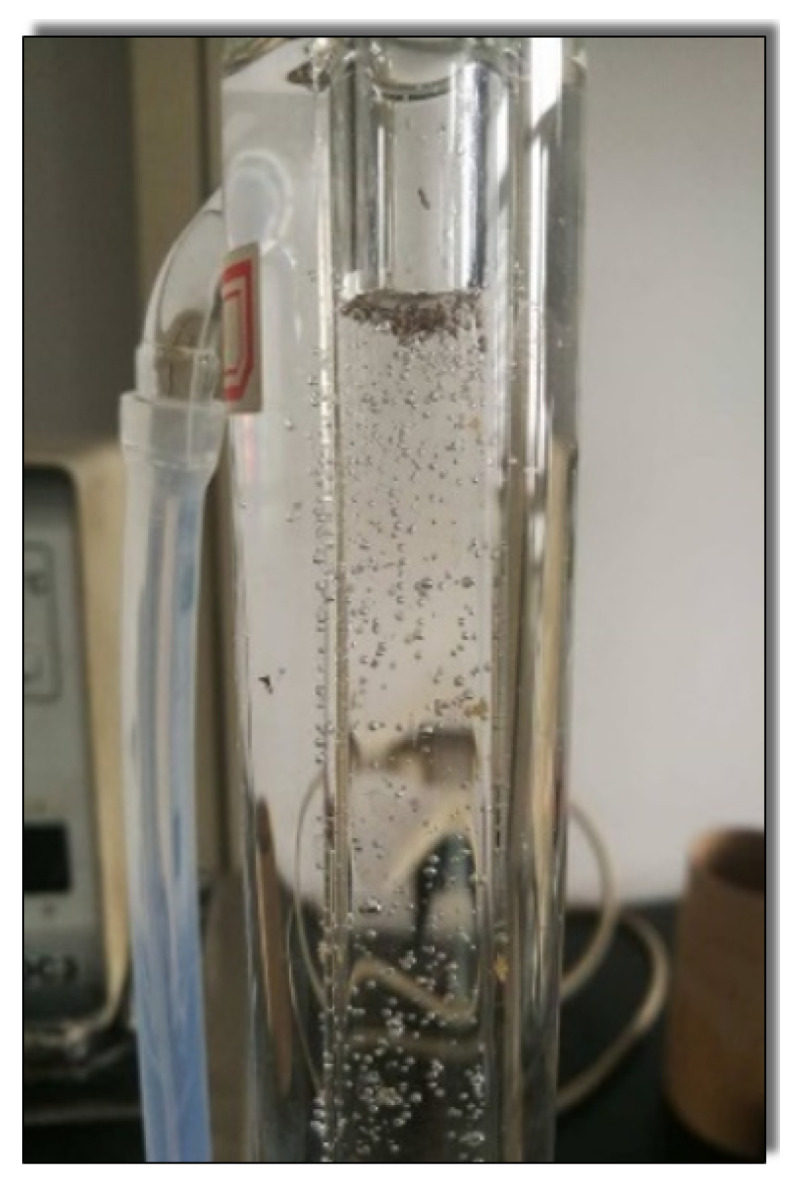
Apparatus setup for sand suspension ability test.

**Figure 6 polymers-12-01470-f006:**
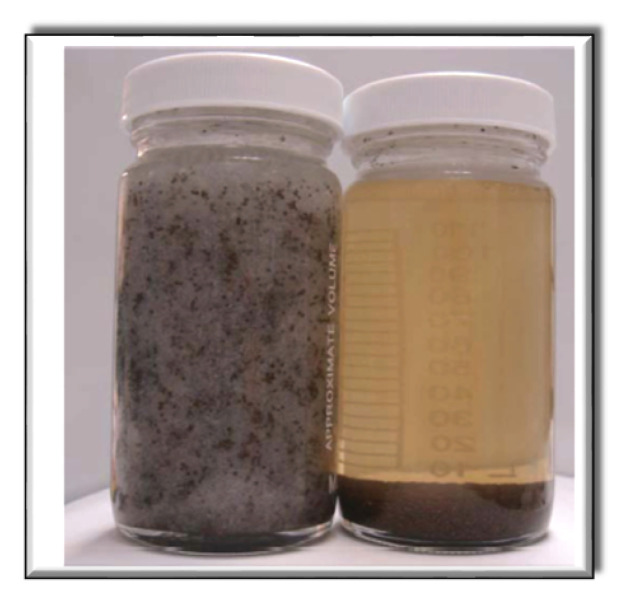
Sand suspension test’s solutions.

**Figure 7 polymers-12-01470-f007:**
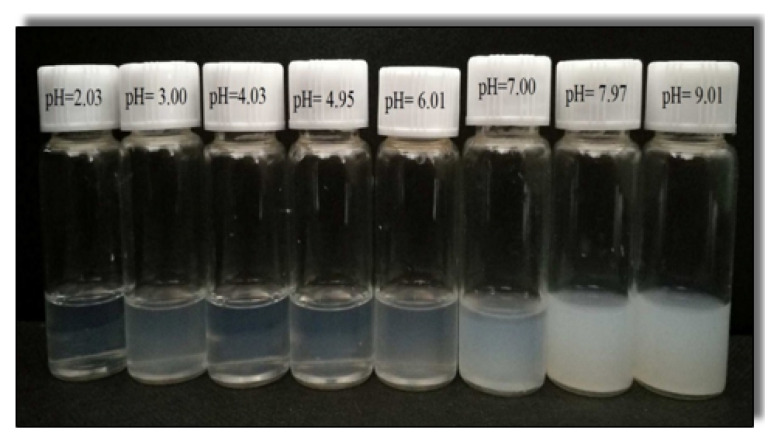
Phase behaviour of CTAB-citric acid (CTAB-CA) solution at different pH values.

**Figure 8 polymers-12-01470-f008:**
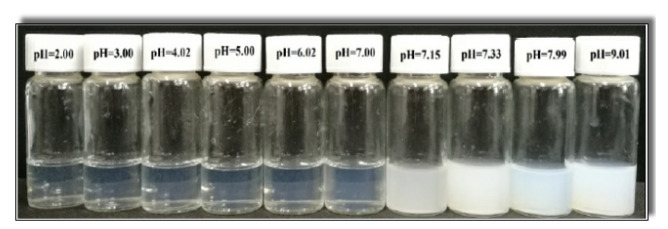
Phase behaviour of CTAB-MA solution at different pH values.

**Figure 9 polymers-12-01470-f009:**
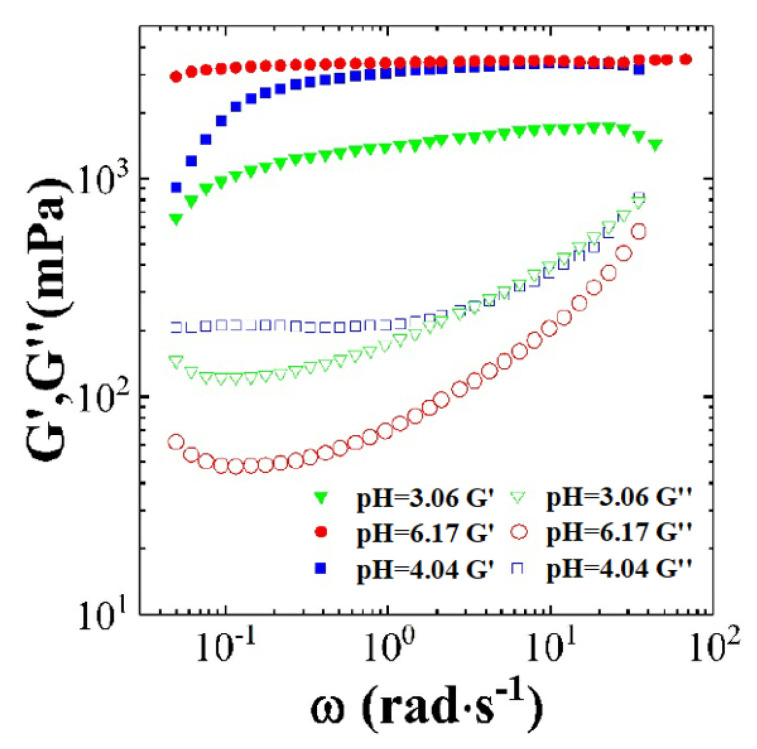
The dynamic rheological curves for the CTAB-CA based (30 mM CTAB + 15 mM CA) VES-fluid at pH range (3 < pH < 7) and at 25 ${}^{\circ }$C.

**Figure 10 polymers-12-01470-f010:**
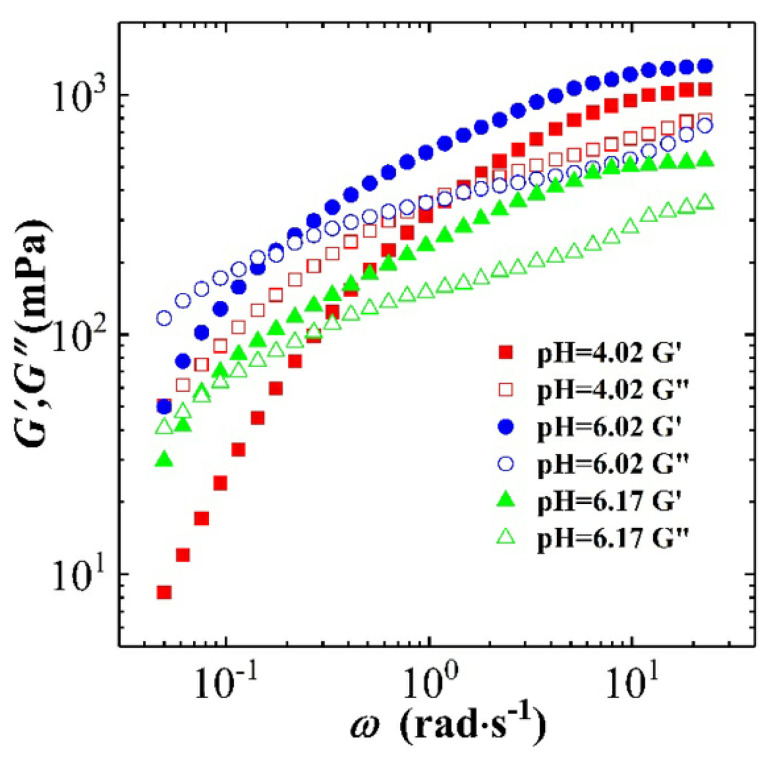
The dynamic rheological curves for the CTAB-MA based (30 mM CTAB + 15 mM MA) VES-fluid at pH range (4 < pH < 7) and at 25 ${}^{\circ }$C.

**Figure 11 polymers-12-01470-f011:**
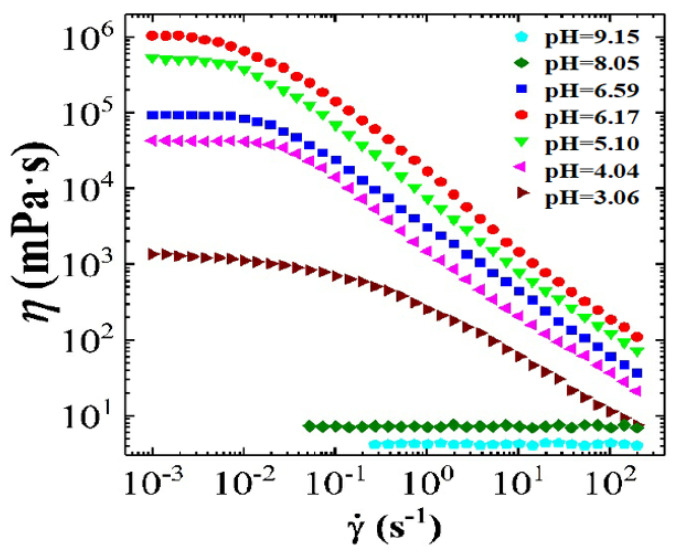
The steady rheological curves for the CTAB-CA based (30 mM CTAB + 15 mM MA) VES-fluid at pH range (3 < pH < 10) and at 25 ${}^{\circ }$C.

**Figure 12 polymers-12-01470-f012:**
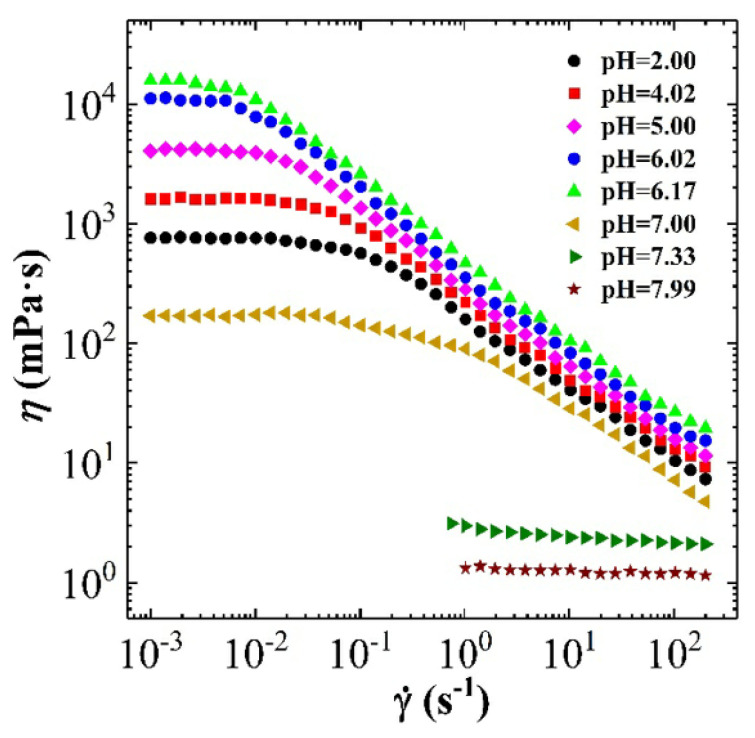
The steady rheological curves for the CTAB-MA based (30 mM CTAB + 15 mM MA) VES-fluid at pH range (2 ≤ pH < 8) and at 25 ${}^{\circ }$C.

**Figure 13 polymers-12-01470-f013:**
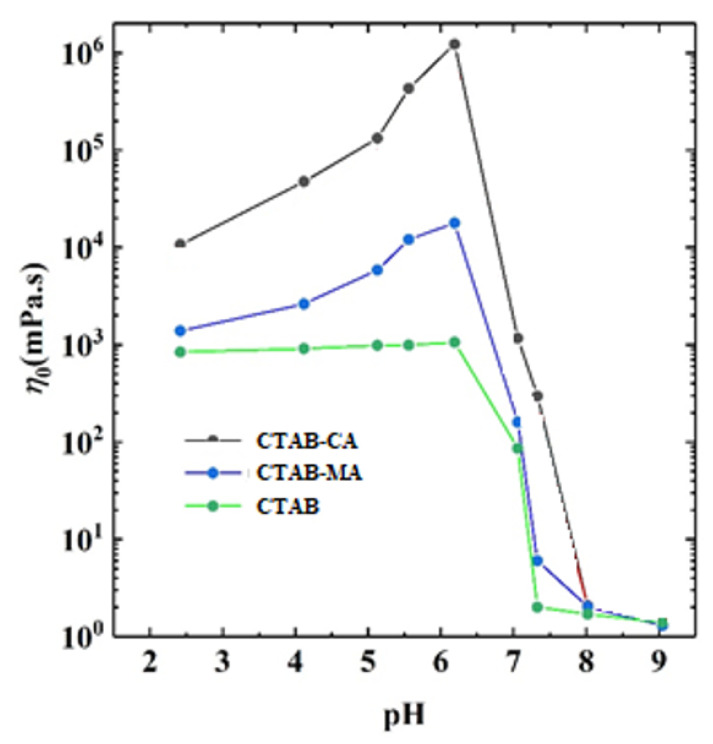
The apparent viscosity for the three types of CTAB based (CTAB, CTAB-MA, and CTAB-CA) VES-fluid at different pH-values, at constant low shear rate, and at 25 ${}^{\circ }$C.

**Figure 14 polymers-12-01470-f014:**
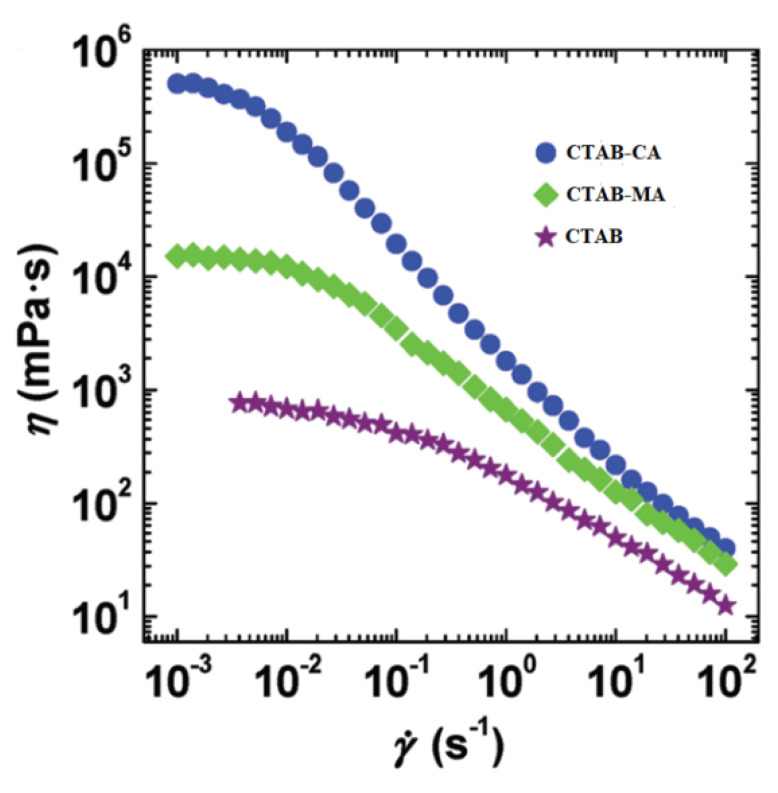
The apparent viscosity for the three types of CTAB based (CTAB, CTAB-MA, and CTAB-CA) VES-fluid at different shear rate, and at 25 ${}^{\circ }$C.

**Figure 15 polymers-12-01470-f015:**
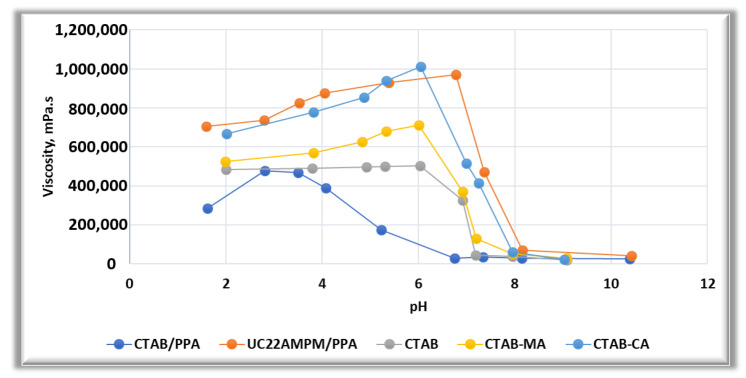
The effect of pH on the apparent viscosity for the three types of CTAB based (CTAB, CTAB-MA, and CTAB-CA) VES-fluids when compared to the literature’s data of Wang et al. [43,63].

**Figure 16 polymers-12-01470-f016:**
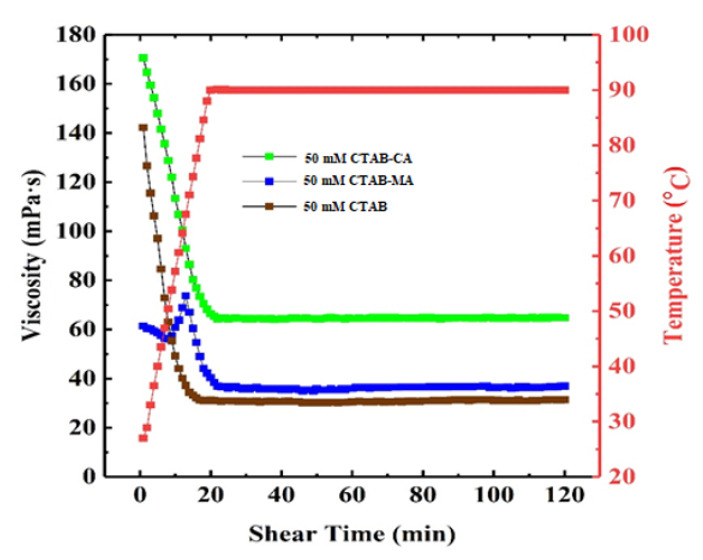
Effect of temperature and alternate shear rate time on the apparent viscosity of VES fluids at shear rate of 170 s${}^{-1}$.

**Figure 17 polymers-12-01470-f017:**
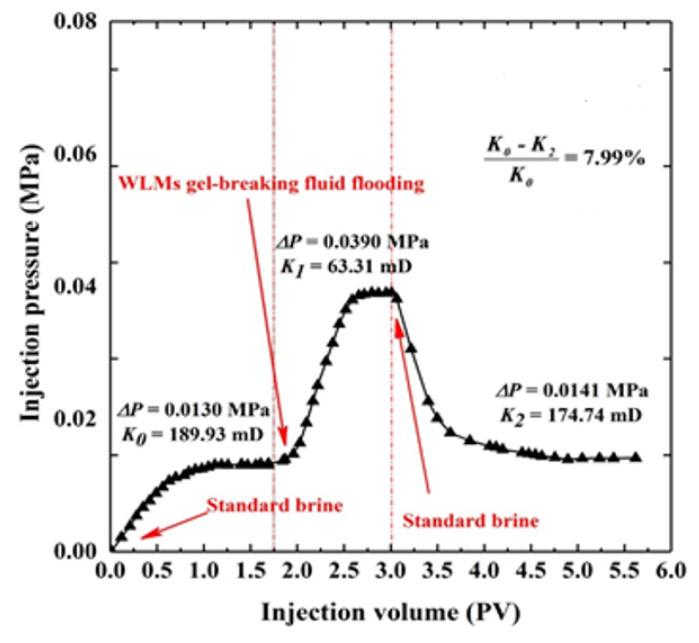
The effect of temperature and alternate shear rate time on the apparent viscosity of VES fluids at shear rate of 170 (s${}^{-1}$).

**Table 1 polymers-12-01470-t001:** Average sand suspension velocity of all VES fluids under different temperatures.

VES-Fluid	Temperature [C]	Height [mm]	Average Suspension Time [s]	Average Suspension Time [mm/s]
30 mM CTAB	30	240	-	-
30 mM CTAB	90	240	54.5	4.37
30 mM CTAB-MA	30	240	-	-
30 mM CTAB-MA	70	240	103.4	2.32
30 mM CTAB-CA	30	240	-	-
30 mM CTAB-CA	70	240	143.6	1.67

**Table 2 polymers-12-01470-t002:** Summary of the breaking properties of different VES fluids at ethanol to fluid ratio of 15:100 and different temperatures.

VES-Fluid	Ethanol to Fluid Ratio	Breaking Temperature [C]	Breaking Time [min]	Viscosity after Breaking [mPa.s]
30 mM CTAB	15:100	30	60	3.4
30 mM CTAB	15:100	80	20	-
30 mM CTAB-MA	15:100	30	152	2.5
30 mM CTAB-MA	15:100	80	60	4.4
30 mM CTAB-CA	15:100	30	120	3.9
30 mM CTAB-CA	15:100	80	-	3.2

**Table 3 polymers-12-01470-t003:** The surface tension and interfacial tension of different VES fluids at ethanol to fluid ratio of 15:100 and different temperatures.

VES-Fluid	Breaking Temperature [C]	Surface Tension [mN/m]	Viscosity after Breaking [mPa.s]
30 mM CTAB	30	27.4	0.4373
30 mM CTAB	80	27.4	0.4373
30 mM CTAB-MA	30	29.1	0.518
30 mM CTAB-MA	80	29.1	0.518
30 mM CTAB-CA	30	23.6	0.414
30 mM CTAB-CA	80	23.6	0.414

**Table 4 polymers-12-01470-t004:** Core properties before and after core flooding.

Condition	Length [mm]	Diameter [mm]	Dry Weight [g]	Porosity [%]	Perm [mD]	Pore Volume [cc]	Bulk Volume
Before VES Fluid Flooding	75.8	38.1	187.4	17.4	189.9	14.8	85.1
After VES fluid flooding	75.8	38.1	182.9	19.9	174.7	16.6	87.3

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
