# Peer review of "Experimental Investigation and Performance Evaluation of Modified Viscoelastic Surfactant (VES) as a New Thickening Fracturing Fluid"

_polymers, 2020, doi:10.3390/polym12071470_

Round 1

Reviewer 1 Report

Dear authors,

Your manuscript introduces idea of utilizing a gemini-like surfactant based on CTAB as a thickener for fracturing fluids. Although you have fairly good results but the manuscript has various scientific mistakes and unfortunately there is strong lack for the scientific explanation for the results. For instance, you could not explain the reason of transition from Newtonian to non-Newtonian behavior with the pH, in terms of structural changes (Fig. 15). You should first describe the equilibrium microstructure at different pH (frequency sweep test Fig 16) then discuss Fig. 15. In Fig 16, you attributed this frequency sweep to that of WLMs. I do not think that WLMs can show such curve of strong gel-like rheograms. 

Also, you could not explain the difference between the effect of CA and MA on the microstructure, and why CTAB-CA shows gel-like behavior while CTAB-MA show a behavior similar to Maxwell behavior.

Reviewer 2 Report

The viscoelastic surfactant is a thickener for fracturing fluid is interesting topic.

The present study presented some novel data. However, some parts have to be improved for publication in Polymers.

Detailed comments on the paper as bellows;

1. Abstract is too long while the novelty is not clearly established.

2. Graphical abstract didnot cover all content of this study.

3. The author should avoid to cite many references in one sentences, 1-4, 12-15, 24-29,...

4. The introduction of surfactant is too less. The characteristics of surfactant should be introduced. 

The way of writing should be improved.

Please refer some references:

Journal of Molecular Liquids 287, 110900; Materials 12 (3), 450

5. For the oil and gas industry application, which kinds of organic substances are widely used

6. Why  did the author only study cationic surfactant, CTAB while many surfactants are widely used.

7. Figure 2 and 3 should be combined in one.  Figs 4 and 5 should combine in one Figure.

8. Figures 6 and 7, 9 and 10, 11 and 12 are the photos of apparatus that are not necessary. The author should remove these photos.

9. Why did the rheological graphs did not match with the photos of phase behavior CTAB-CA

10. The rheological curves for CTAB+MA also didnot match with the photo of phase behavior CTAB-MA

11. The temperature and shear resistance should be in more details.

12.Conclusions are too long but the new findings have not been emphasized again.

Round 2

Reviewer 2 Report

The revised manuscript was significantly improved by the authors.

The responses to reviewers' comments are also suitable.

I think the paper can be accepted for publication in Polymers.